# Calcium Supplementation in Pregnancy: A Systematic Review of Clinical Studies

**DOI:** 10.3390/medicina61071195

**Published:** 2025-06-30

**Authors:** Angeliki Gerede, Panayiota Papasozomenou, Sofoklis Stavros, Anastasios Potiris, Ekaterini Domali, Nikolaos Nikolettos, Makarios Eleftheriades, Menelaos Zafrakas

**Affiliations:** 1Department of Obstetrics and Gynecology, Democritus University of Thrace, 69100 Alexandroupolis, Greece; nnikolet@med.duth.gr; 2School of Health Science, International Hellenic University, 57400 Thessaloniki, Greece; ppapasoz@ihu.gr; 3Department of Obstetrics and Gynecology, University General Hospital “ATTIKON”, Medical School, 9 National and Kapodistrian University of Athens, 12462 Athens, Greece; sfstavrou@med.uoa.gr (S.S.); apotiris@med.uoa.gr (A.P.); 4Department of Obstetrics and Gynecology, Alexandra Hospital, Medical School, National and Kapodistrian University of Athens, 11528 Athens, Greece; kdomali@yahoo.fr; 5Department of Obstetrics and Gynecology, University Hospital Aretaieion, Medical School, National and Kapodistrian University of Athens, 11528 Athens, Greece; melefth@med.uoa.gr

**Keywords:** calcium supplementation, calcium intake, preeclampsia, gestational hypertension

## Abstract

*Background and Objectives*: Calcium is an essential mineral that plays a vital role in fetal development and maternal health during pregnancy. The World Health Organization recommends a daily calcium intake of 1.5–2 g for pregnant adult women. Calcium deficiency during gestation may lead to complications, such as gestational hypertension, preeclampsia, loss of bone mineral density, impaired fetal development, and other adverse pregnancy outcomes. The aim of the present review is to evaluate the current clinical evidence on calcium intake during pregnancy. *Methods*: The present systematic review was conducted according to the PRISMA 2020 statement by searching two major databases, PubMed and Mendeley. The study protocol was registered in the Open Science Framework (DOI: osf.io/rvj7z). Inclusion criteria were clinical trials on calcium supplementation during pregnancy. Exclusion criteria were clinical guidelines, reviews, case reports, case series, letters, and commentaries. The Newcastle–Ottawa Scale was used to assess the risk of bias in the included studies. *Results*: Initially, 451 publications were identified, and after removal of duplicates and screening of titles and/or abstracts and/or full texts, 34 studies were included. The number of participants ranged between 30 and 22,000 women. Calcium supplementation was associated with lower incidence of and less severe gestational hypertension and preeclampsia, lower risk of preterm birth, longer pregnancy duration and higher neonatal birth weight, and improved maternal bone mineral density postpartum. When the doses were split up into smaller doses, the benefits were strongest with high-dose regimens (1.5–2 g/day). *Conclusions*: Calcium supplementation during pregnancy has beneficial effects on maternal and neonatal health, especially in populations with insufficient dietary daily calcium intake and women at high risk of hypertensive disorders. Daily dose may vary according to individual needs, daily dietary calcium intake, and general health status. Further large-scale randomized controlled trials (RCTs) are necessary to confirm these findings.

## 1. Introduction

Calcium (Ca) is the most abundant mineral in the human body, accounting for approximately 2% of the total body weight; furthermore, 99% of calcium is stored in the skeletal system. The remaining 1% is distributed in intra- and extracellular fluids, where it is essential for various regulatory functions, including muscle contraction, enzyme and hormone activity, and the maintenance of bone integrity [1,2,3]. Consequently, ionic calcium (Ca^2+^) must be tightly regulated in order to maintain plasma concentrations within normal ranges [4], i.e., 8.5–10.5 mg/dL or 2.2–2.7 mmol/L. Calcium levels are influenced by gastrointestinal absorption, renal excretion, and mobilization from the skeleton, which serves as the body’s primary calcium reservoir [5].

During pregnancy, the maternal body undergoes significant physiological adaptations in order to maintain homeostasis and support fetal growth and development. These adaptations affect calcium metabolism directly, as calcium plays a critical role in early embryonic stages, particularly in cell signaling and control of cell division, as well as in the third trimester by supporting fetal skeletal mineralization [6,7]. Transfer of calcium through the placenta increases as pregnancy progresses, and this often results in a negative calcium balance for the mother [8,9]. Even women with adequate calcium intake early in pregnancy may experience a neutral or negative balance during the third trimester due to heightened fetal bone accretion demands [10,11]. The developing fetus requires between 50 and 330 mg of calcium per day for skeletal development, with demands increasing significantly as pregnancy progresses [12,13]. Insufficient maternal calcium intake may increase the risk of gestational complications, including hypertensive disorders, preterm delivery, and long-term maternal morbidities, such as excessive bone loss [10].

The association between calcium intake and pregnancy-induced hypertension (PIH) has been studied extensively since the 1980s. Early research on calcium metabolism in pregnancy dates back to 1934, when maternal hypocalcemia during late gestation was observed [14,15]. The hypothesis of an inverse association between calcium intake and PIH emerged in 1980, following observations of low hypertensive disorder rates among Mayan women in Guatemala, whose traditional preparation of corn using lime water (calcium hydroxide) resulted in high dietary calcium intake [16]. Subsequent studies have underscored the importance of adequate calcium intake during pregnancy, elucidating mechanisms of calcium homeostasis and its impact on maternal and fetal health [17,18,19,20]. Insufficient calcium intake during pregnancy is associated with adverse outcomes for both mother and fetus, including hypertensive disorders, impaired bone mineralization, muscle cramps, preterm birth, low birth weight, and poor fetal bone mineralization [21,22]. Hypertensive disorders of pregnancy include gestational hypertension, chronic hypertension, preeclampsia (PE), and eclampsia [23,24,25,26]. PE affects 3–7% of nulliparous women and 1–3% of multiparous women and remains a leading cause of maternal mortality worldwide [21,27].

It has been estimated that in 2011, more than 3 billion people were at risk of calcium deficiency, primarily in Africa and Asia [28]. In low- and middle-income countries and at least 27% of high-income countries, average calcium intake is below 800 mg/day [1]. Additionally, more than 50% of pregnant women in the United States fail to meet the recommended dietary calcium intake [29]. To address this issue, the World Health Organization (WHO) recommends calcium supplementation at doses of 1.5–2 g/day for pregnant women with insufficient dietary calcium intake [2].

Calcium supplementation represents a cost-effective strategyto mitigate the adverse outcomes of calcium deficiency during pregnancy, including PIH, preterm birth, and maternal morbidity, particularly in populations with low dietary calcium intake [30]. Combining supplementation with calcium-rich or calcium-fortified foods is critical in order to achieve adequate total calcium intake [3,31]. However, high doses of calcium in a single administration may have adverse effects. Therefore, calcium supplementation should be prescribed in divided doses throughout the day in order to enhance absorption, minimize side effects, and improve tolerability and effectiveness [2]. Despite the widespread recommendation by the WHO for calcium supplementation (1.5–2 g/day) during pregnancy in populations with low dietary intake [2], implementation and clinical practice vary across the globe. Previous systematic reviews and meta-analyses [13,14] have highlighted the potential of calcium supplementation in reducing the incidence of hypertensive disorders and improving perinatal outcomes. However, inconsistencies persist regarding optimal dosage, timing, supplement formulation, and efficacy in diverse populations. These knowledge gaps, along with rising global concern about maternal hypertensive disorders and inadequate calcium intake in both low- and high-income settings [1,15], justify an updated and comprehensive synthesis of the available clinical evidence. The aim of the present systematic review was to evaluate the role of calcium intake during pregnancy and the indications for calcium supplementation and to identify significant factors that may affect effective absorption of calcium, including dosage regimen, supplement formulation, routine administration, and prescription practices.

## 2. Materials and Methods

For this systematic review, a literature search was conducted by using two major databases: PubMed and Mendeley. The study protocol was registered in the Open Science Framework (DOI: osf.io/rvj7z). The search terms to identify relevant studies were “(calcium) AND (intake OR supplementation) AND (pregnancy OR gestation) AND (clinical trials)”. The search algorithm was adjusted for each database while maintaining a common overall rationale. Publications concerning calcium supplementation and pregnancy published until 22 November 2024 were included. The following inclusion criteria were used: clinical trials on calcium supplementation during pregnancy. The following exclusion criteria were set: clinical guidelines, any type of review article, case reports and case series, letters, and commentaries.

For transparent and reproducible reporting of this systematic review, the PRISMA 2020 (Preferred Reporting Items for Systematic Reviews and Meta-Analyses) statement was used [32]. The PRISMA checklists for the Abstract and the main text are provided in Appendix A, respectively. After the initial literature search, two independent authors (A.G. and P.P.) screened the articles for relevance based on the titles and abstracts. Disagreements were resolved through consensus or discussion with a third author (S.S.). Articles deemed irrelevant were excluded, and the full-text copies of the remaining articles were assessed for eligibility by two independent reviewers (A.G. and A.P.). Inconsistencies were resolved through consensus or by a third reviewer (E.D.). The references of the full-text copies were accessed to prevent the potential loss of eligible studies that might have been missed by the database search (snowball procedure). The following data items were extracted from the eligible studies: year of publication, study design, country, center and time period during which the study was conducted, number of participants, age, sample size, population characteristics (e.g., geographic location, demographic data), intervention details (e.g., calcium dosage, supplementation duration), comparators, and primary and secondary outcomes.

## 3. Results

### 3.1. Study Selection Process

Figure 1 presents a flow diagram of the search and selection process. A total of 451 publications were initially identified. After removal of duplicates, 433 articles were considered eligible for title and abstract screening. Of these, 393 articles were excluded according to the eligibility criteria. Consequently, 40 studies were sought for retrieval, and 38 studies were retrieved. Eight studies were excluded for the following reasons: animal studies (*n* = 3), full text was missing (*n* = 1), and calcium was contained in multivitamin preparations (*n* = 4). Additionally, references from the included studies and other relevant studies published in high-impact journals were hand-searched, leading to the inclusion of four more papers. Thus, a total of 34 studies were included investigating the association between calcium supplementation and maternal or fetal/neonatal outcomes. In detail, 14 studies evaluated the association between calcium intake and PIH, 11 studies assessed the incidence, risk, and/or severity of PE, 7 studies explored the association between calcium and preterm birth, 4 studies investigated serum calcium levels, and 3 studies evaluated the association with systolic/diastolic blood pressure (BP) and birth weight. Only two studies focused on fetal growth or bone mineral density (BMD). A summary of the characteristics of the included studies, along with their most important findings, is provided in Table 1.

### 3.2. Gestational Calcium Supplementation Outcomes

As pregnant women undergo physiological changes to support fetal development, calcium homeostasis is disrupted [7]. The fetus depends on maternal calcium sources, and its needs are particularly pronounced in the third trimester, when fetal skeletal mineralization peaks [10,29]. It has been estimated that the fetus receives approximately 50 mg of calcium per day at 20 weeks, and this amount increases to 330 mg per day by 35 weeks, in order to support skeletal development [7,8,13]. Maternal calcium absorption is directly influenced by dietary calcium intake [63]. The most well-studied adverse effects of this disruption in pregnant women are hypertensive disorders, changes in bone density and skeletal health, and adverse neonatal outcomes.

#### 3.2.1. Hypertensive Disorders

Six clinical studies evaluated the impact of calcium supplementation on systolic and diastolic BP during gestation [18,36,48,49,51,59]. Studies conducted after 20 weeks’ gestation in healthy women consistently reported lower systolic and diastolic BP values in the groups receiving calcium supplementation. These findings suggest a positive effect of calcium supplementation on BP regulation, with no difference observed with different dosage levels [18,36,49,51]. In contrast, calcium supplementation had no significant effect in women with mild PE and nulliparous women before 20 weeks [48,59].

Fourteen clinical studies investigated the association between calcium intake and PIH. Ten of these studies reported a significant positive effect of calcium supplementation [18,22,39,47,49,50,55,56,58,61]. Women at high risk for PE and adolescents with low dietary intake or with a history of PE benefited the most from high-dose calcium supplementation [22,47,58,61]. No benefit was observed in women with prior PE who received gradually increased doses of calcium or who already had high dietary calcium intake [20,40,60]. One study reported no effect of high-dose calcium supplementation on PIH incidence in healthy nulliparous women [53]. Other studies reported delayed onset [57] and decreased severity of PIH in the groups receiving calcium supplementation [22].

#### 3.2.2. Preeclampsia (PE)

Among pregnancy complications related to calcium supplementation, PE is the most extensively studied. Seventeen studies included in the present review evaluated the effects of calcium supplementation on PE risk, incidence, and severity [18,20,22,38,40,42,46,51,52,53,54,55,56,57,59,61,62]. A positive effect was reported in 13 studies. In contrast, there was no significant impact in three studies on women who had already experienced PE [20,54,59] and in one study in which supplementation was discontinued before the third trimester [53]. In most studies, positive outcomes were associated with high-dose calcium supplementation (>1.2 g/day); notably, there were two studies that reported comparable benefits with low-dose supplementation (0.5–0.6 g/day) [42,46]. One study suggested that high-dose calcium supplementation (1.5 g/day) divided into three doses was more effective in preventing PE than a single daily dose [57]. With early calcium supplementation initiated before 20 weeks’ gestation, the severity of PE was significantly reduced in normotensive primigravidas, although the impact on PE incidence was lower [62]. In adolescent pregnancies, 2 g of daily calcium supplementation was associated with a 12.35% risk reduction of PE [51].

#### 3.2.3. Preterm Birth

The effect of calcium supplementation on preterm birth, which is often linked to PIH and PE, is a key area of interest. Seven clinical studies in the present review investigated the incidence of preterm birth or pregnancy duration in relation to calcium supplementation. Results from all of these trials indicated that calcium supplementation could reduce preterm birth rates or extend pregnancy duration [22,40,42,50,52,55,61]. However, a study conducted simultaneously in India and Tanzania reported no impact of calcium supplementation in the Indian cohort [42]. Besides reducing preterm birth rates, one study also demonstrated a significant reduction of intrauterine growth restriction rates among women who received calcium supplementation [52].

#### 3.2.4. Maternal Bone Mineral Density (BMD) and Skeletal Health

In this systematic review, two clinical studies evaluated the association between maternal BMD and calcium supplementation during pregnancy [19,41]. The first study was conducted in the United States between 2011 and 2013 and involved 64 healthy women with a mean daily calcium intake of 0.7 g. Participants received 1 g of calcium supplementation daily from 16 weeks’ gestation to delivery. Results showed that women who received calcium supplementation had a 4–5% greater BMD during the first 12 months postpartum compared with those who did not [19]. The second study was conducted in Brazil in adolescent pregnant women with low dietary calcium intake and provided participants with 0.6 g of daily calcium supplementation and vitamin D starting at 26 weeks’ gestation. The group that received supplementation exhibited higher lumbar spine bone mass and a reduced rate of femoral neck bone loss during lactation [41].

#### 3.2.5. Birth Weight

Clinical studies evaluating birth weight focused on women at high risk for low birth weight due to age (adolescent women) or hypertensive profiles [20,50,55,61]. In three studies, participants received 2 g of daily calcium supplementation from the third trimester to delivery, while one study began with low-dose calcium supplementation before pregnancy, increasing it to 1.5 g/day in the third trimester [20]. Both adolescent populations and hypertensive high-risk groups who received calcium supplementation of 2 g daily delivered babies with significantly higher birth weights compared with the control groups [50,55,61].

#### 3.2.6. Infant Skeletal Growth and Bone Mineralization

A limited number of studies investigated the association between maternal calcium supplementation and infant skeletal growth. Two recent studies found no significant differences in skeletal growth between the group receiving calcium supplementation and the control group [33,34]. However, another study in pregnant women with low dietary calcium intake (0.6 g/day) reported a 15% increase in fetal bone mineralization following 2 g of daily calcium supplementation initiated before 20 weeks’ gestation compared with the control group [43]. Bone mineralization was assessed using dual-energy X-ray absorptiometry (DEXA), which is considered the gold standard for BMD in both maternal and fetal studies. Furthermore, a study involving bedridden mothers receiving low-dose calcium supplementation (0.5 g/day) demonstrated an improved calcium status in their preterm neonates [45]. The assessment was carried out by measuring the serum calcium levels and through quantitative ultrasound (QUS), a non-invasive tool that is used to evaluate bone health in neonates when DEXA is impractical.

#### 3.2.7. Other Pregnancy Outcomes

Clinical studies examining the impact of calcium supplementation on maternal outcomes, such as cardiac pulse and cholesterol levels, reported no significant effects [35,37]. In contrast, one study of pregnant women exposed to high environmental lead levels showed that high doses of calcium supplementation, initiated in the first trimester, were associated with significantly reduced lead levels in blood [44].

### 3.3. Calcium Dose

In most studies included in the present review, high doses of calcium supplementation were compared with placebos. Only four studies compared low and high calcium doses. In two studies, women in the study group received 1.5–2 g daily (high dosage), while women in the control group received 0.5 g of calcium supplementation daily (low dosage); with 2 g of daily calcium, a significant reduction in the incidence of PE and preterm birth was noted [52]. In a study conducted in India and Tanzania, results were different between the two countries, as a benefit from high rather than low calcium dosage in reducing preterm births was demonstrated only in the cohort from Tanzania [42]. The timing and frequency of calcium intake were investigated in a study conducted in Indonesia involving 140 women who received 1.5 g of calcium daily; women who received calcium supplementation divided into three doses had a significantly lower risk of PE compared with those who took only a single dose [57].

### 3.4. Risk of Bias Assessment

A formal quality assessment was performed for all 34 included studies by using the Newcastle–Ottawa Scale (NOS) [64]. Most studies had a moderate to low risk of bias. The overall quality of evidence was deemed sufficient to support the review’s conclusions. A full summary of NOS scoring is provided in Table 2.

Most of the studies had a low risk of bias (score ≥ 7) at 22 out of 34 studies (65%), which were mostly RCTs with unambiguous randomization, blinding, and adjusted confounders (e.g., [18,19]). The other studies showed a moderate risk of bias (scoring 5–6). In total, 11 out of 34 studies (32%) had this score because the blinding was not clear ordietary calcium intake was not fully controlled (e.g., [39,44]). Finally, one study [54] had high risk of bias (score ≤ 4) because it did not have a placebo control and not all outcomes were reported.

## 4. Discussion

Calcium homeostasis during pregnancy is essential for pregnancy progression and favorable maternal and neonatal outcomes. During pregnancy, maternal intestinal calcium absorption is gradually increased in order to meet the additional fetal demand. Women receiving adequate calcium intake (>1.0 g/day) absorbed 57% during the second trimester and 72% of calcium during the third trimester [10]. However, some investigators suggested that women in pregnancy do not require an increase in calcium intake, as the physiological adaptive processes during this period are independent of maternal calcium intake [3]. Similarly, other studies support the view that women who already meet the recommended dietary calcium intake (>1.0 g/day) may not need additional calcium [11,30,43].

In 2019, a study showed that 75% of the women in the USA consume daily calcium up to 40% below the recommended dietary allowance [19]. In 2022, another study noted that in most low-income and middle-income countries, calcium intake during pregnancy is suboptimal, and there are high rates of mortality due to maternal hypertensive disorders [28]. In 2019, another group reported that high-income countries also face this issue, with 27% of women consuming insufficient calcium during pregnancy [1]. Pregnant women with inadequate daily calcium intake due to various health problems (including chronic auto-immune disorders and lactose intolerance or those receiving daily low-molecular-weightheparin therapy) or pregnant women who do not consume milk and dairy products due to personal preference may be at higher risk of gestational hypertensive disorders, poor bone mineralization, and higher morbidity and mortality rates. Several studies support the benefits of calcium supplementation during pregnancy for improving maternal and neonate outcomes [1,10,13,18,19,22,28,35,36,38,39,40,42,43,44,45,46,47,49,50,51,52,53,54,55,56,57,58,61,62,65].

Gestational hypertensive disorders constitute the leading cause of maternal and perinatal morbidity and mortality, with especially high rates in developing countries [66,67,68,69]. PIH, PE, and eclampsia are the most common hypertensive disorders during pregnancy [23,24]. It has been estimated that approximately 5–10% of women experience high BP during pregnancy [7,25,26]. Two independent studies in Cameroon showed even higher rates, with hypertensive disorders reaching up to 21.2% of maternal deaths in the period between 2017 and 2019 [70,71]. Effective strategies for preventing PE include anti-platelet agents, aspirin, and calcium supplementation [8,72]. Calcium supplementation appears to be the most cost-efficient and safe intervention in order to reduce the incidence of PE and gestational hypertension [30,73] and in consideration of the associated health care costs when administered to all pregnant women instead of subgroups only [74]. A recent meta-analysis that focused on PE and included 26 RCTs concluded that calcium supplementation reduced the risk of PE by 49% (OR 0.51, 95% CI 0.41–0.64), although there was a lot of heterogeneity (I^2^ = 72%) [73]. The conclusions of the present systematic review are in line with this meta-analysis; in addition, it should be emphasized that the benefits are greatest for high-risk populations, such as adolescents and those with low dietary intake. Furthermore, our conclusions are in line with those of the WHO 2013 recommendation of 1.5–2 g/day calcium for PE prevention in low-intake populations, which was based on meta-analyses showing a 52% reduction in PE risk (RR 0.48, 95% CI 0.33–0.69) and a 30% lower PIH risk (RR 0.70, 95% CI 0.57–0.86) [2].

In 1991, a study suggested that pregnant women who received calcium supplementation after the 20th week of pregnancy had a lower risk of PIH [18]. Later, in 1993, a study that included subgroups with different daily calcium doses ranging from 0.12 g to 2 g concluded that the highest dose of 2 g of daily calcium supplementation may reduce the incidence of PIH [39]. In contrast to these findings, a study held in India and Tanzania involving 11,000 women in each country reported that there was no difference with high calcium doses regarding the risk of PE [42]. Consistently, there were similar findings in most studies included in the present review with high-dose calcium supplementation in the last trimester [18,22,39,46,47,49,50,51,52,56,58,61], as well as with lower doses (0.5–1.0 g/day) when there was no pre-existing PE [35,36,41,45,46,47]. Groups that benefit the most from calcium supplementation include adolescents and women at high risk for hypertensive disorders [46,51,55,58,61]. In contrast, in the case of pre-pregnancy supplementation or early termination of calcium supplementation before the third trimester, there were minimal or no benefits at all [20,38,40,51,62].

Pre-pregnancy calcium supplementation had neither a positive nor a negative effect on the incidence of new-onset hypertension in primiparas or the incidence of recurrent PE in parous women [20,75]. Overall, most clinical studies conclude that calcium supplementation during pregnancy may reduce the risk of PE by 49–52%, the risk of PIH by 30%, and the risk of severe PE by 25% [73]. However, others caution that focusing solely on the mean of random-effects meta-analyses for PE, without accounting for substantial heterogeneity, may be misleading [76].

In several studies, the impact of calcium supplementation on BP alterations was compared to that of control groups. Calcium supplementation initiated after 20 weeks’ gestation in healthy, normotensive women demonstrated positive effects, with reductions observed in both systolic and diastolic BP, regardless of dosage [18,36,49,51]. In contrast, studies in hospitalized women with mild PE or in cases where supplementation was discontinued before 20 weeks reported no significant impact [48,59]. The precise mechanism throughwhich calcium supplementation lowers BP remains unclear, as no significant changes in cardiac output or other laboratory parameters were observed [37,48].

Beyond hypertensive disorders, calcium is essential for bone formation and bone turnover, with the skeleton serving as the primary reservoir of calcium in the human body. When serum calcium levels drop, calcium is mobilized from bones to maintain homeostasis. Active placental calcium transport supports the rapidly mineralizing fetal skeleton, but this may increase maternal bone resorption and reduce maternal BMD, particularly when dietary calcium intake is inadequate [7]. Two studies included in the present review found that calcium supplementation during pregnancy may increase BMD and improve postpartum bone recovery, reducing the long-term risk of osteoporosis in women [19,41]. Furthermore, other studies have shown that increased bone resorption during pregnancy, combined with low dietary calcium intake, is associated with lower maternal BMD for up to five years postpartum. These findings support the recommendation for antenatal calcium supplementation in order to promote lifelong maternal bone health [77], particularly for adolescents, who are still in the process of completing their own bone mineralization and are at higher risk of calcium deficiency due to the dual demands of their own growth and pregnancy [41,51]. However, other studies have shown limited benefits from calcium supplementation. One study [78] reported no significant improvement in maternal or offspring BMD with calcium supplementation; another study [79] highlighted a potentially increased risk of hip fractures and bone loss associated with high calcium intake. Other researchers, however, warned that bone resorption markers may stay high for years, indicating that calcium supplementation may need to be continued after pregnancy [77].

Maternal calcium supplementation of up to 2 g/day during the second and third trimesters has been shown to enhance fetal bone mineralization in women with low dietary calcium intake [43]. Daily supplementation of 1.2 g has been associated with a 15% increase in fetal bone mineral content, highlighting the critical role of calcium in supporting fetal skeletal health [43]. In one study, 2 g/day of calcium supplementation was associated with increased fetal bone mineralization by 15% [43]. On the other hand, a 2022 meta-analysis found no long-term differences in the BMD of the offspring [78]. Additionally, improved calcium intake has been linked to better neonatal outcomes in bedridden mothers, as preterm neonates exhibited enhanced calcium status when their mothers received 0.5 g/day of calcium supplementation [45].

Calcium supplementation during pregnancy appears to reduce the risk of preterm birth, likely due to its role in lowering parathyroid hormone (PTH) release and intracellular calcium levels, which in turn decrease uterine smooth muscle contractility [64,68]. A 2012 meta-analysis that included 13 RCTs linked calcium supplementation to a 24% lower preterm birth risk (RR 0.76, 95% CI 0.60–0.97) [69]. Furthermore, longer pregnancy duration associated with calcium supplementation has been linked to increased neonatal birth weight, with gains ranging from 85 to 552 g compared with placebo groups [50,55,61,69]. However, similarly to findings related to hypertensive disorders, calcium supplementation in women with prior PE has no significant impact on the risk of preterm birth or on birth weight [20].

Calcium supplementation is generally safe and effective for pregnant women. Daily supplementation of 0.5–1.0 g of calcium is recommended to achieve daily uptake of at least 1.0 g of calcium [13]. WHO guidelines recommend 1.5–2.0 g/day of calcium supplementation during pregnancy [2]. However, implementation is often hindered by cost, logistical challenges, and gastrointestinal discomfort. Regular antenatal visits and partner involvement appear to improve adherence [65]. The two most commonly used oral calcium formulations are calcium carbonate and calcium citrate. Calcium carbonate is cost-effective and can be taken alongside iron, while calcium citrate is preferred for those women who have gastric issues [28,29]. Large supplementation doses (1.5–2 g calcium/day) may cause gastrointestinal discomfort, including bloating and heartburn [29]. For better absorption and avoidance of these effects, doses of ≤0.5 g are recommended, taken at mealtimes [2,28]. For women with heartburn, calcium-based antacids offer both symptom relief and supplementation [29]. For high-risk populations, a food-based approach incorporating calcium-rich sources alongside supplementation remains the optimal strategy for ensuring bone health [28]. A recent study from 2023 [57] found that divided doses (e.g., 500 mg thrice daily) enhanced efficacy for PE prevention versus a single dose. Excessive calcium intake (> ~2500mg/day) has been linked to cardiovascular risks, emphasizing the importance of adhering to recommended doses [80]. Finally, calcium supplementation should be prescribed cautiously, tailored to individual risks and benefits, in order to maximize safety and efficacy in preventing maternal and neonatal complications [81].

In the present review, clinical studies combining calcium with vitamin D [35,36,41,54] or linoleic acid [46,47] were intentionally included and carefully considered for a number of reasons. First, in prenatal care, calcium is often given in combination with vitamin D in order to improve absorption or with linoleic acid in order to reduce inflammation in hypertensive disorders. Clinically significant data would be missed if these studies were excluded. Second, the WHO recognizes that, in practice, calcium and vitamin D supplements are frequently taken together, particularly in populations that are deficient in both nutrients [2]. Furthermore, studies combining calcium with vitamin D or linoleic acid were included because adjuncts were uniformly administered to both the intervention and control groups, and the primary outcome (e.g., PE risk) was tested against a placebo. In fact, we excluded those trials in which calcium was part of a multivitamin supplement (*n* = 4 excluded, per Methods) to avoid uncontrolled confounding.

The main strength of the present systematic review is that it provides a structured overview of clinical studies on calcium supplementation during pregnancy in association with multiple maternal and neonatal outcomes. Limitations of this review include the fact that only two databases were searched, small sample sizes in many of the included primary studies limit the reliability and external validity of results, and inconsistencies in defining dietary calcium intake, supplementation timing, and serum calcium levels preclude comparisons between studies. The heterogeneity in study populations and sampling times, along with a lack of racial–ethnic diversity, may have also influenced the findings.

## 5. Conclusions

Overall, calcium supplementation averaging 1.2 g/day from mid-pregnancy to term has been shown to mitigate hypertensive disorders, improve fetal bone mineral content, reduce postpartum maternal bone loss, and increase pregnancy duration and birth weight. These protective measures are most beneficial for women at highest risk, such as adolescents, those with low calcium intake, and women with a history of PE or low dietary intake. Dosing, supplement formulation, and prescription timing are very important in order to maximize absorption and manage adverse effects. High-dose regimens (1.5–2 g/day) are better absorbed when divided into doses of ≤0.5 g. However, it is important that more high-quality, high-powered clinical studies with greater racial–ethnic diversity and more homogenous design with respect to dosing and genetic background are conducted.

## Figures and Tables

**Figure 1 medicina-61-01195-f001:**
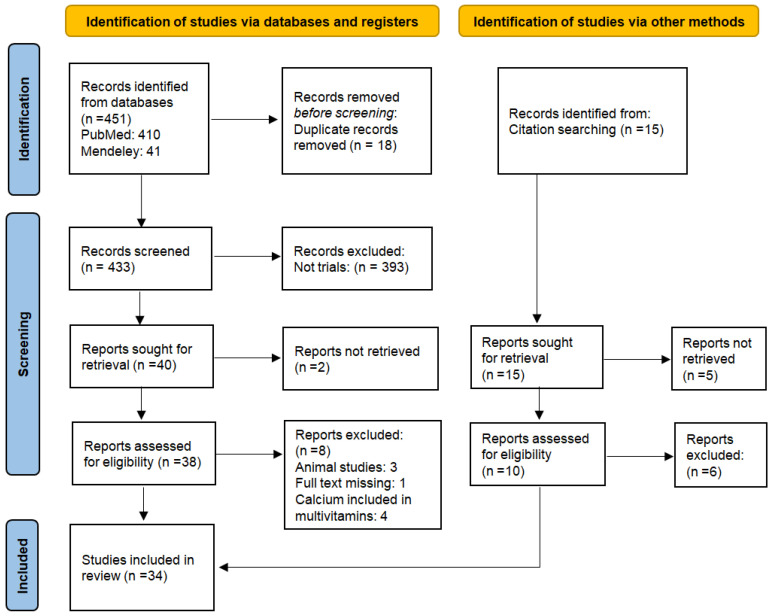
PRISMA flow chart of the included studies.

**Table 1 medicina-61-01195-t001:** Summary of characteristics of the included studies.

Study	Population Characteristics	Study Group (Ca^2+^/Day)	Calcium Form	Control	Key	Main Findings (In Study Group)
Abalos et al. [33]	Argentina (*n* = 510 primiparous/low Ca^2+^ intake)	1.5 g Ca^2+^/d (<20 w)	CCA	Placebo	FG	No impact on fetal somatic or skeletal growth.
Abdel-Aleem et al. [34]	Argentina, Egypt, India, Peru, South Africa, Vietnam (*n* = 91/low Ca^2+^ intake)	1.5 g Ca^2+^/d (<20 w)	CCA	Placebo	FG	No impact on fetal and infant growth during 1st year.
Asemi et al. [35]	Iran (*n* = 49 singletons/18–35 years old)	0.5 g Ca^2+^+ 200IU Vit.D/d (>25 w)	CCI	Placebo	Other	Decrease in FPG, serum triglycerides, and total cholesterol. No impact on HDL/LDL.
Asemi et al. [36]	Iran (*n* = 42/18–40 years old)	0.5 g Ca^2+^+200IU Vit.D/d (>25 w)	CCI	Placebo	S/D	Increased serum Ca^2+^. Decreased diastolic pressure. No impact on pregnancy outcome.
Belizán et al. [18]	Argentina (*n* = 1194 nulliparous)	2 g Ca^2+^/d (>20 w)	CCA	Placebo	S/D, PIH PE	Decreased systolic and diastolic pressure. Decreased risk of PIH and PE.
Boggess et al. [37]	USA (*n* = 18 healthy pregnant)	1.5 g Ca^2+^/d (28–31 w)	CCA	Placebo	Other	No impact on cardiac pulse.
Carroli et al. [38]	Argentina (*n* = 510 nulliparous/low Ca^2+^ intake)	1.5 g Ca^2+^/d divided into 3 doses (<20 w)	CCA	Placebo	PE	Reduced severity of PE complications.
Cong et al. [39]	China (*n* = 318 primiparas)	Ca^2+^ (g)/d 0.12 g, 0.24 g, 1 g, 2 g (20 w)	CCA	Placebo	PE	Decreased risk of PIH in 2 g Ca^2+^/d group.
Crowther et al. [40]	Australia (*n* = 459 singleton nulliparous, high intake)	1.8 g Ca^2+^ (<24 w)	CCA	Placebo	PIH, PE PB	Decreased risk of PE and preterm birth. No impact on PIH.
Cullers et al. [19]	USA (*n* = 64)	1 g Ca^2+^/d (>16 w)	CCI	Placebo	BMD	Improved bone recovery postpartum. Increased BMD.
Diogenes et al. [41]	Brazil (*n* = 56)	0.6 g Ca^2+^ + Vit.D (>26 w)	CCI	Placebo	BMD	Increased BMD.
Dwarkanath et al. [42]	India (*n* = 11,000) Tanzania (*n* = 11,000)	1.5 g Ca^2+^/d	CCA	0.5 g Ca^2+^/d	PE PB	No inferiority of high dose supplementation for risk of PE. Decreased risk of preterm live birth only in Tanzania high-dose group.
Koo et al. [43]	USA (*n* = 256)	2 g Ca^2+^/d (<20 w)	CCA	Placebo	BMD	Enhanced fetal bone mineralization in women with low calcium intake.
Ettinger et al. [44]	Mexico (*n* = 670 exposed to lead)	1.2 g Ca^2+^ (1st trimester to delivery)	CCI	Placebo	Other	Decreased blood lead levels.
Gioxari et al. [45]	Greece (*n* = 42 bedridden mothers + 42 preterm neonates)	0.5 g Ca^2+^/d	CCI	-	BMD	Enhanced calcium status in preterm neonates of bedridden mothers receiving calcium.
Herrera et al. [46]	Colombia (*n* = 86/primigravidas HR)	600 mg Ca^2+^ + linoleic acid/d (>24 w)	CCA	Placebo	PE	Significant decrease in the incidence of PE.
Herrera et al. [47]	Colombia (*n* = 48/HR healthy primigravidas with family history of hypertension)	600 mg Ca^2+^ + linoleic acid/d (>18 w)	CCA	Placebo	PIH	Decreased PIH.
Hofmeyr et al. [48]	South Africa (*n* = 708 nulliparous)	1.5 g Ca^2+^/d (<20 w)	CCA	Placebo	S/D	No effect on the rate of abnormal laboratory measures associated with PE.
Hofmeyr et al. [20]	S. Africa, Zimbabwe, Argentina (*n* =1355 parous with previous PE)	0.5 g Ca^2+^/d (<20 w) and 1.5 g Ca^2+^/d (>20 w)	CCA	Placebo	PE, PB PIH, BW	No statistical significant impact.
López-Jaramillo et al. [49]	Ecuador (*n* = 106 nulliparous)	2 g Ca^2+^/d (>24 w)	CCA	Placebo	S/D PIH	Decreased systolic and diastolic BP. Decreased risk of PIH.
López-Jaramillo et al. [50]	Ecuador (*n* = 56) PIH HR	2 g Ca^2+^/d (>28 w)	CCA	Placebo	PIH, PB BW	Decreased incidence of PIH. Increased duration of pregnancy and mean birth weight.
López-Jaramillo et al. [51]	Ecuador (*n* = 260/<17,5 years old)	2 g Ca^2+^/d (>20 w)	CCA	Placebo	S/D, PE	Decreased systolic and diastolic BP. Decreased risk of PE (12.35%).
Khan et al. [52]	Developing countries (*n* = 272)	2 g Ca^2+^/d (>20 w)	CCA	0.5 g Ca^2+^/d (>20 w)	PE, PB	High-dose daily calcium reduced PE incidence, preterm birth, and IUGR.
Levine et al. [53]	USA (*n* = 4589/nulliparous)	2 g Ca^2+^/d (13–21 w)	CCA	Placebo	PIH PE	No impact on PE or PIH risk. No impact on perinatal outcomes or adolescent pregnancy outcomes.
Marya et al. [54]	India (*n* = 400 toxaemic)	375 g Ca^2+^+1.2IU VitD/d	CLA	-	PE	No impact.
Niromanesh et al. [55]	Iran (*n* = 30/HR)	2 g Ca^2+^/d	CCA	-	PE, PIH, BW	Decreased risk of PE. Delayed onset of PIH (3 w). Longer duration of pregnancy. Increased infant weight (mean 552 g).
Purwar et al. [56]	India (*n* = 201 nulliparous)	2 g Ca^2+^/d (>20 w)	CCA	Placebo	PIH PE	Decreased risk of PIH and PE.
Qurniyawati et al. [57]	Indonesia (*n* = 140)	1.5 g Ca^2+^/d divided in 3 doses	CCA	1.5 g Ca^2+^/d in <3 doses	PE	Decreased risk of PE when Ca^2+^ supplementation divided into 3 doses within the day.
Sanchez-Ramos et al. [58]	USA (*n* = 281/HR nulliparous)	2 g Ca^2+^/d (>24–28 w)	CCA	Placebo	PIH	Decreased incidence of PIH.
Sanchez-Ramos et al. [59]	USA (*n* = 75 hospitalized due to mild PE)	2 g Ca^2+^/d (24–3 w)	CCA	Placebo	PE S/D	No impact on S/D pressure orprevention of severe PE in patients with mild disease.
Rogers et al. [60]	China (*n* = 500 normotensive primi-gravidas)	0.6 g Ca^2+^/day (22–32 w) 1.2 g Ca^2+^/day (>32 w)	CCA	Placebo	PIH	No impact of Ca^+2^ in reducing the incidence of PIH.
Villar et al. [61]	USA (*n* = 190/<17 years old)	2 g Ca^2+^/d (>23 w)	CCA	Placebo	PB, BW PIH, PE	Decreased incidence of preterm labor and birth. Decreased incidence of PIH and PE. Increased birth weight and mean duration of labor.
Villar et al. [22]	Argentina, Egypt, India, Peru, and South Africa (*n* = 8325 low dietary calcium intake <0.6 g/d)	1.5 g Ca^2+^ (>20 w)	CCA	Placebo	PE, PIH PB	No impact on PE incidence; decreased severity of PE, PIH, maternal morbidity, and neonatal mortality. In women <20 years, decreased incidence of preterm and early preterm delivery.
Wanchu et al. [62]	India (*n* = 100 normotensive primi-gravidas)	2 g Ca^2+^/d (<20 w)	CCA	-	PE	No impact on PE incidence. Decreased severity of PE.

BMD = bone mineral density; BW = birth weight; d = day; CCA= calcium carbonate; CCI = calcium citrate; CLA = calcium lactate; FG = fetal growth; FPG = Fasting Plasma Glucose; HR = high risk for PE; IUGR = intrauterine growth restriction; low Ca^2+^ intake = <600 mg/day; PB = preterm birth; PIH = pregnancy-induced hypertension; PE = preeclampsia; S/D = systolic/diastolic pressure; w = week(s).

**Table 2 medicina-61-01195-t002:** Risk of bias assessment of all included clinical studies by using the Newcastle–Ottawa Scale (NOS).

Study	Selection (Max 4)	Comparability (Max 2)	Outcome (Max 3)	Total Score (Max 9)
Abalos et al. [33]	★★★	★★	★★	7
Abdel-Aleem et al. [34]	★★★	★	★★	6
Asemi et al. [35]	★★★★	★★	★★★	9
Asemi et al. [36]	★★★★	★★	★★★	9
Belizán et al. [18]	★★★★	★★	★★★	9
Boggess et al. [37]	★★★	★	★★	6
Carroli et al. [38]	★★★★	★★	★★★	9
Cong et al. [39]	★★★	★	★★	6
Crowther et al. [40]	★★★★	★★	★★★	9
Cullers et al. [19]	★★★★	★★	★★★	9
Diogenes et al. [42]	★★★	★★	★★	7
Dwarkanath et al. [42]	★★★★	★★	★★★	9
Koo et al. [43]	★★★★	★★	★★★	9
Ettinger et al. [44]	★★★	★	★★	6
Gioxari et al. [45]	★★★	★★	★★★	6
Herrera et al. [46]	★★★★	★★	★★★	9
Herrera et al. [47]	★★★★	★★	★★★	9
Hofmeyr et al. [48]	★★★★	★★	★★★	9
Hofmeyr et al. [20]	★★★★	★★	★★★	9
López-Jaramillo et al. [49]	★★★★	★★	★★★	9
López-Jaramillo et al. [50]	★★★★	★★	★★★	9
López-Jaramillo et al. [51]	★★★★	★★	★★★	9
Khan et al. [52]	★★★	★	★★	6
Levine et al. [53]	★★★★	★★	★★★	9
Marya et al. [54]	★★	★	★	4
Niromanesh et al. [55]	★★★	★	★★	6
Purwar et al. [56]	★★★	★★	★★★	7
Qurniyawati et al. [57]	★★★	★	★★	6
Sanchez-Ramos et al. [58]	★★★★	★★	★★★	9
Sanchez-Ramos et al. [59]	★★★	★	★★	6
Rogers et al. [60]	★★★	★	★★	6
Villar et al. [61]	★★★★	★★	★★★	9
Villar et al. [22]	★★★★	★★	★★★	9
Wanchu et al. [62]	★★★	★	★★	6

## Data Availability

The original data presented in the study are openly available on Medline/PubMed (https://pubmed.ncbi.nlm.nih.gov/) and Mendeley (https://www.mendeley.com/datasets) accessed on 22 November 2024.

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
