# Peer review of "Calcium Supplementation in Pregnancy: A Systematic Review of Clinical Studies"

_medicina, 2025, doi:10.3390/medicina61071195_

Round 1
Reviewer 1 Report
Comments and Suggestions for Authors
The systematic review provided a comprehensive overview of clinical studies regarding calcium supplementation during pregnancy and its association with various maternal and neonatal outcomes. While the review is well-written, several modifications are necessary.
- The rationale for the study should be examined before outlining the objectives, emphasizing the existing gaps in the data, and citing relevant references from previous systematic reviews.
- Please remove the names of the authors from the discussion, such as Cullers et al., Cormick et al., and Belizán et al. Additionally, please rephrase the sentences accordingly.
- Line 266, in 2000 Prentice suggested should be revised without the name of the author.
- There are too many instances of blood pressure and Preeclampsia in the text and tables. Please write out blood pressure (BP) and Preeclampsia (PE) in the first use and write only the abbreviations "BP” and “PE” for the remaining words.
- The background and objective of the abstract are too long. Please summarize this section to allow for more findings in the results.
- The objective stated in the abstract does not align with the conclusion. Please revise it accordingly.
- In the discussion, please compare the results with those of other previous systematic reviews and meta-analyses.
Proofreading is required.
Author Response
RESPONSES TO REVIEWER 1
- Reviewer 1 Comment 1: The rationale for the study should be examined before outlining the objectives, emphasizing the existing gaps in the data, and citing relevant references from previous systematic reviews.
Authors’ Response: In order to provide a clearer explanation of the rationale behind this systematic review, we have updated the introduction section accordingly. In detail, the high prevalence of low calcium intake in pregnant women worldwide, especially in low- and middle-income nations, is highlighted, and the corresponding elevated associated risks, such as preterm birth, gestational hypertension, preeclampsia, and impaired fetal development, are described as well. We reported that there are still disagreements about various aspects, including, among others, the best dosage, timing, formulation, and population-specific advantages of calcium supplementation during pregnancy, even after earlier research and systematic reviews.
- Reviewer 1 Comment 2: Please remove the names of the authors from the discussion, such as Cullers et al., Cormick et al., and Belizán et al. Additionally, please rephrase the sentences accordingly.
Authors’ Response: We have removed the authors’ names and rephrased accordingly. We have also done the same with Nicolas et al, Gomez et al., Wright et al, Tihtonen et al., and Warensjö et al.
- Reviewer 1 Comment 3: Line 266, in 2000 Prentice suggested should be revised without the name of the author.
Authors’ Response: We have removed the author’s name and rephrased accordingly.
- Reviewer 1 Comment 4: There are too many instances of blood pressure and Preeclampsia in the text and tables. Please write out blood pressure (BP) and Preeclampsia (PE) in the first use and write only the abbreviations "BP” and “PE” for the remaining words.
Authors’ Response: We have used the abbreviations for the remaining words.
- Reviewer 1 Comment 5: The background and objective of the abstract are too long. Please summarize this section to allow for more findings in the results.
Authors’ Response: We have shortened the “Background/Objective” subsection and extended the “Results”.
- Reviewer 1 Comment 6: The objective stated in the abstract does not align with the conclusion. Please revise it accordingly.
Authors’ Response:We have changed the “Background/Objective” subsection.
- Reviewer 1 Comment 7: In the discussion, please compare the results with those of other previous systematic reviews and meta-analyses.
Authors’ Response:The discussion has been improved in order to include comparisons with other previous systematic reviews and meta-analyses.
Reviewer 2 Report
Comments and Suggestions for Authors
Here are my comments and suggestions to improve this work.
1- Why do the authors use only two databases? Is it possible to add Scopus, CENTRAL, EMBASE, or WoS? Please consider adding almost one of the databases cited before, and the systematic review becomes more comprehensive.
2- In lines 48 and 49, please insert the range of ionic calcium in the plasma.
3- In Table 1 please insert a column that indicates the type of Calcium form (e.g. calcium carbonate, citrate, calcium lactate etc.)
4- Table 1, why the authors consider supplementation of Calcium plus another supplement for example Vit. D or Linoleic acid, these are confounding factors.
Consider removing the clinical trial using another supplement in addition to calcium.
5- Table 1 Line Marya et al (53), is the calcium dose 375 grams? maybe a typo intending milligrams. And what do you intend with the term toxaemic?
6- In line 213 and 331 use only BMD, the full term is written before.
7- 3.2.6 section, insert the type of tools used for the assessment of bone mineralisation.
e.g. MOC or DEXA
8- line 287 uses only PIH, the full term is present before.
9- Line 375 what is the Calcium dose that increases the cardiovascular risk? Insert the dose.
Author Response
RESPONSES TO REVIEWER 2
- Reviewer 2 Comment 1: Why do the authors use only two databases? Is it possible to add Scopus, CENTRAL, EMBASE, or WoS? Please consider adding almost one of the databases cited before, and the systematic review becomes more comprehensive.
Authors’ Response: Indeed, we used two databases. Adding more databases at this point is not possible, as this would in practice mean that we would have to conduct the entire study from scratch. We have discussed this limitation in the Discussion section.
- Reviewer 2 Comment 2: In lines 48 and 49, please insert the range of ionic calcium in the plasma.
Authors’ Response:We have added the range of ionic calcium in the plasma.
- Reviewer 2 Comment 3: In Table 1 please insert a column that indicates the type of Calcium form (e.g. calcium carbonate, citrate, calcium lactate etc.)
Authors’ Response: This information has been added as an extra (fourth) column.
- Reviewer 2 Comment 4: Table 1, why the authors consider supplementation of Calcium plus another supplement for example Vit. D or Linoleic acid, these are confounding factors. Consider removing the clinical trial using another supplement in addition to calcium.
Authors’ Response: This may be indeed a confounding factor and we have now discussed this issue in the Discussion section.
- Reviewer 2 Comment 5: Table 1 Line Marya et al (53), is the calcium dose 375 grams? maybe a typo intending milligrams. And what do you intend with the term toxaemic?
Authors’ Response: We have corrected this typing error (the correct is 375 mg/day) and “toxaemic” was replaced by “PE” (i.e. preeclampsia).
- Reviewer 2 Comment 6: In line 213 and 331 use only BMD, the full term is written before.
Authors’ Response: We have now used only the acronym BMD.
- Reviewer 2 Comment 7: 2.6 section, insert the type of tools used for the assessment of bone mineralisation. e.g. MOC or DEXA
Authors’ Response:The section has been modified accordingly.
- Reviewer 2 Comment 8: line 287 uses only PIH, the full term is present before.
Authors’ Response:We have now used only the acronym PIH in pages 3, 7, and 9.
- Reviewer 2 Comment 9: Line 375 what is the Calcium dose that increases the cardiovascular risk? Insert the dose.
Authors’ Response:It has been added along with the relative literature.
Round 2
Reviewer 1 Report
Comments and Suggestions for Authors
The authors tried to modify the manuscript. It has the potential to be accepted for publication.
iThenticate report indicates a 24% similarity. Please reduce it to below 19%.
Comments on the Quality of English LanguageModerate proofreading is required.
Reviewer 2 Report
Comments and Suggestions for Authors
The authors responded to my suggestions and integrated the lacking parts.